# Bevacizumab-induced proteinuria and its association with antihypertensive drugs: A retrospective cohort study using a Japanese administrative database

Anna Kiyomi[1]*, Fukiko Koizumi[1], Shinobu Imai[1,2,3], Hayato Yamana[2,4], Hiromasa Horiguchi[2], Kiyohide Fushimi[2,5], Munetoshi Sugiura[1]

1 Department of Drug Safety and Risk Management, School of Pharmacy, Tokyo University of Pharmacy and Life Sciences, Tokyo, Japan, 2 Department of Clinical Data Management and Research, Clinical Research Center, National Hospital Organization Headquarters, Tokyo, Japan, 3 Division of Pharmacoepidemiology, Department of Healthcare and Regulatory Sciences, School of Pharmacy, Showa University, Tokyo, Japan, 4 Data Science Center, Jichi Medical University, Tochigi, Japan, 5 Department of Health Policy and Informatics, Graduate School of Medical and Dental Sciences, Tokyo Medical and Dental University, Tokyo, Japan

* akiyomi@toyaku.ac.jp

**Data Availability Statement:** The third party data for this study contain potentially identifying or

## Abstract

Proteinuria is a major side-effect of the anti-tumor drug bevacizumab, although its incidence and risk factors in the real world are still unclear. Although renin-angiotensin-aldosterone system inhibitors are used clinically to prevent proteinuria, their efficacy remains unclear. The aim of the present study was to reveal the incidence and risk factors of bevacizumab-induced proteinuria and examine the effectiveness of antihypertensive drugs in preventing proteinuria. We conducted a retrospective cohort study using the National Hospital Organization Clinical Data Archives and Medical Information Analysis Databank. Hospitalized patients who received bevacizumab between January 1, 2016, and June 30, 2019, were included. The study outcome was proteinuria within 12 months of bevacizumab administration. Patient characteristics, laboratory tests, and medications were compared between patients with and without proteinuria using multivariable logistic regression analysis. Among the 2,458 patients, 27% developed proteinuria after bevacizumab administration. Nursing dependence (odds ratio [OR], 2.40; 95% confidence interval [CI], 1.89–3.05; P<0.001) and systolic blood pressure ≥140 mmHg (OR, 1.44; 95% CI, 1.17–1.79; P<0.001) were identified as risk factors. Patients with an estimated glomerular filtration rate (eGFR) of 60–89, 45–59, and <45 mL/min/1.73 m$^2$ had 29.7%, 76.8%, and 66.0% higher odds of proteinuria, respectively, than those with an eGFR ≥90 mL/min/1.73 m$^2$. No significant relationship was observed between antihypertensive drugs and the occurrence of proteinuria. More patients may suffer from proteinuria after bevacizumab administration than previously reported. Nursing dependence and systolic blood pressure are predictive risk factors for bevacizumab-induced proteinuria. Patients at risk of proteinuria should be closely monitored.

**Funding:** This work was supported by JSPS KAKENHI Grant Number 21K17258. The funder had no role in study, design, data collection and analysis, decision to publish, or preparation of the manuscript.

**Competing interests:** The authors have declared that no competing interests exist.

# Introduction

Bevacizumab, a vascular endothelial growth factor inhibitor, is widely administered for the treatment of colorectal, non-small-cell lung, breast, and ovarian cancers [1–4]. Hypertension and proteinuria are the major side effects of bevacizumab. According to the package insert, the incidence of proteinuria is 10.4%, although a meta-analysis of 16 randomized controlled trials reported an incidence of 13.3% [5]. However, the actual incidence and onset time of bevacizumab-induced proteinuria remains unclear.

Antihypertensive drugs have been empirically administered to prevent and treat bevacizumab-induced proteinuria [6]. Renin-angiotensin-aldosterone system (RAA) inhibitors (including angiotensin receptor blockers [ARBs] and angiotensin-converting enzyme inhibitors [ACEIs]), which increase nitric oxide levels, have been reported to prevent and treat bevacizumab-induced proteinuria [7]. However, bevacizumab indirectly inhibits endothelial nitric oxide synthase, lowering nitric oxide levels [8, 9]. Thus, RAA inhibitors may antagonize bevacizumab activity and lower its efficacy [10]. Moreover, bevacizumab-induced proteinuria has been studied in limited populations [6, 7, 10], and a real-world investigation on the appropriate antihypertensive drug for preventing/treating bevacizumab-induced proteinuria is required.

Real-world evidence is a valuable tool for assessing drug effectiveness and safety, especially among patients who are often excluded from randomized controlled trials [11]. Such evidence, using daily healthcare service provisions, laboratory data, and patient medical records, reflects the actual effect and safety of the drug in patients [11].

The present study investigated the risk factors of bevacizumab-induced proteinuria and examined the effectiveness of antihypertensive drugs in preventing proteinuria using a Japanese nationwide administrative database.

# Materials and methods

## Study design and patient selection

This retrospective cohort study used data from the National Hospital Organization (NHO) in Japan. All hospitalized patients who were administered bevacizumab between January 1, 2016, and June 30, 2019, were included. The index day (day 1) was defined as the first day of bevacizumab administration within the inclusion period. The exclusion criteria were as follows: receipt of bevacizumab within six months prior to day 1; age <20 years; an outlier value for weight (<25 kg) or height (<100 cm) (individuals with such values were considered to have been registered incorrectly, for example, as 0, or had switched weight and height values); proteinuria on or before day 1; lack of laboratory test results pertaining to the period after day 1; and receipt of dialysis within 12 months after day 1.

This study was approved by the Institutional Review Board of Tokyo University of Pharmacy and Life Sciences (Approval No. 2019–020) and the NHO (Approval No. R1-1219001). The requirement for informed consent was waived because of the anonymous nature of the data. This research was conducted in accordance with the Strengthening the Reporting of Observational Studies in Epidemiology (STROBE) statement.

## Data sources

The NHO covers more than eight million patients in 140 hospitals throughout Japan. It provides two databases: the NHO Clinical Data Archives (NCDA), which collects real-time clinical information from the electronic medical records of NHO hospitals, and the Medical Information Analysis (MIA) data bank, which collects daily insurance claims information

[12]. Sixty-six NHO hospitals and all NHO hospitals record NCDA and MIA data, respectively. Vital signs and laboratory data were obtained from the NCDA. The MIA includes data for the Diagnosis Procedure Combination/Per-Diem Payment System. This database includes patient information on demographics, primary diagnosis, comorbidities present on admission, complications occurring during hospitalization, medical procedures, medications, and materials; it also contains patient details including age, sex, body height, weight, grade of activities of daily life on admission, discharge converted to the Barthel index [13], nursing dependency score, and discharge status. International Statistical Classification of Diseases and Related Health Problems, 10th Revision (ICD-10), codes (S1 Table) were used to group cancers.

**Outcome measurement and variables.** The outcome of this study was an event of proteinuria within 12 months of the index day. Proteinuria positivity was defined as qualitative proteinuria $\geq$1+, $\geq$10 mg/dL spot urine protein, or $\geq$120 mg/day protein in 24-h urine collection samples. The definition was according to the Common Terminology Criteria for Adverse Events (CTCAE) v5.0. The proteinuria grade was determined based on the result of the first positive qualitative value according to the CTCAE v5.0: grade 1: 1+; grade 2: 2–3+, and grade 3: 4+.

The antihypertensive agent used on the index day was defined by the presence of one of the following [14] (Fig 1): 1) the medication was prescribed on the index day, 2) the days supplied by the previous prescription covered the index day, or 3) the sum of the days supplied between day -27 and day 0 was more than 28.

Patient characteristics included the following (S2 Table): age, sex, weight, height, body mass index (BMI), nursing dependency score (rolling over, transferring, oral hygiene, dietary intake, changing clothes, communication on treatment, and risk behavior associated with patient performance status [15] on the index day), Charlson comorbidity index (CCI) excluding cancer [16, 17], other comorbidities (congestive heart failure, diabetes, or kidney disease), cancer type based on all diagnoses, previous chemotherapy use, chemotherapy (L01A: alkylating-based, L01B: antimetabolites-based, L01C: plant alkaloid-based, L01D: cytotoxic antibiotics-based, L01E: protein kinase inhibitor-based, L01X: platinum-based, and L02B: hormone antagonist-

1. Prescribed on the index day

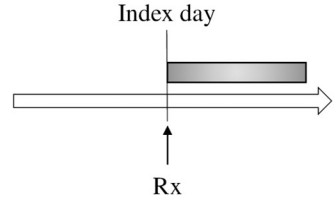

2. The days covered by the previous prescription extended beyond the index day

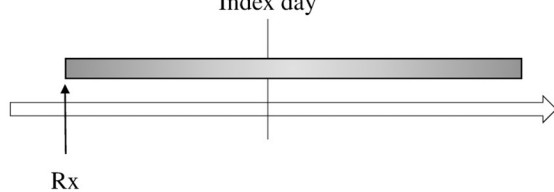

3. The total number of days covered by prescriptions between day -27 to day 0 was more than 28 days

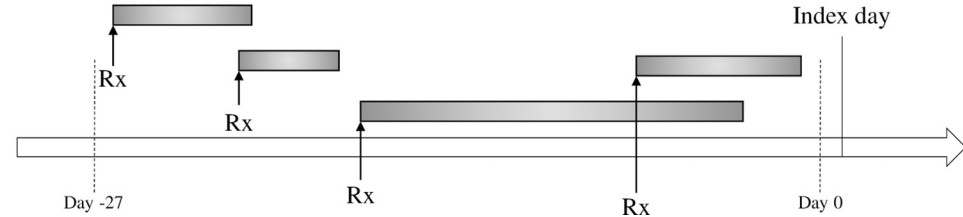

**Fig 1. Definition of antihypertensive drug utilization.** Abbreviated symbols: Rx, prescribed. Each gray bar represents possession of the medication.

based, in the Anatomical Therapeutic Chemical Classification by World Health Organization), and hypertensive treatment on the index day (ARB, ACEI, calcium channel blocker, loop diuretics, thiazide, beta-blockers, spironolactone, and nitrates). The following vital signs and laboratory data measured on the index day or one day prior were used as variables: systolic blood pressure (SBP), diastolic blood pressure (DBP), serum creatinine (Scr), estimated glomerular filtration rate (eGFR), creatinine clearance (Ccr), albumin, blood urea nitrogen, total bilirubin, and glycated hemoglobin (HbA1c). eGFR and Ccr were calculated using the Scr value and categorized based on the Evidence-Based Practice Guideline for the Treatment of Chronic Kidney Disease 2018 released by the Japanese Society of Nephrology. If the Scr value was missing, it was calculated by the Modification of Diet in Renal Disease equation for Japanese patients [18], assuming a GFR of 75 mL/min/1.73 m$^2$ [19].

## Statistical analysis

Patient characteristics were described using the mean and standard deviation for continuous variables and proportions for categorical variables. Predictor variables were compared between patients with and without proteinuria using the chi-square test for categorical variables and Wilcoxon rank-sum test for continuous variables. Multivariate logistic regression analysis was performed to determine the factors associated with the occurrence of proteinuria using the forced entry method. Variables with a P <0.15 were determined using univariate analysis, and BMI, obesity, and antihypertensive drugs were selected for the analysis. Additionally, we compared chemotherapy and antihypertensive drug utilization between patients with and without proteinuria using the chi-square test. Statistical analyses were performed using JMP Pro 15 (SAS Institute Inc., Cary, North Carolina, USA), and P-values less than 0.05 indicated significant differences.

## Results

### Patient identification and the incidence rate of proteinuria

Fig 2 shows the flow of the participant selection process. In total, 3,134 patients were administered bevacizumab during the study period. After exclusions, 2,458 patients were eligible for

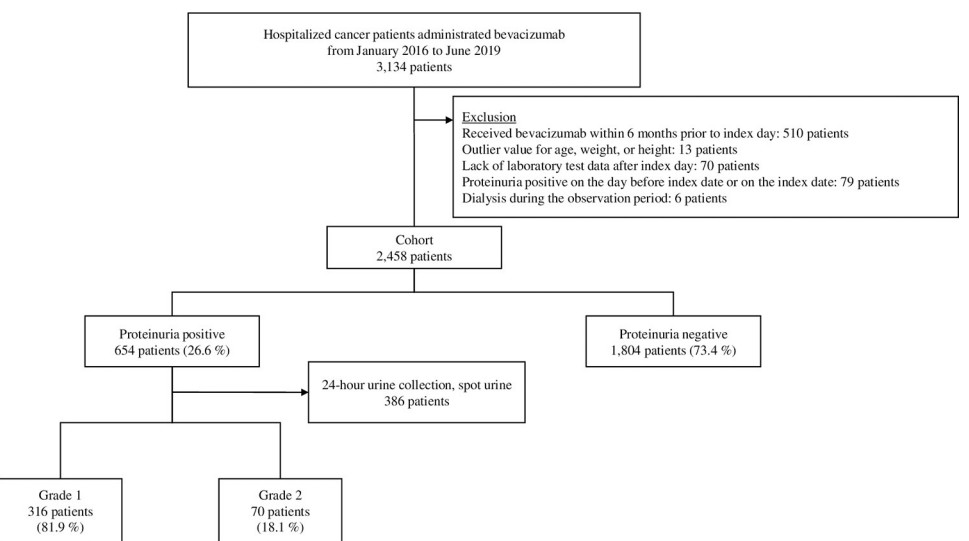

**Fig 2. Characteristics of the excluded and eligible patients in the analysis dataset.**

analysis. Of these patients, 654 (26.6%) experienced an episode of proteinuria within 12 months after the index day, while 1,804 (73.4%) did not experience such an episode. After the exclusion of 386 patients who were diagnosed based on a 24-h urine collection or spot urine quantitative test, 316 (81.9%) and 70 (18.1%) patients exhibited grade 1 and grade 2 proteinuria, respectively. None of the patients exhibited grade 3 proteinuria.

## Patient demographics and clinical characteristics

The patient demographics of those with proteinuria (n = 654) and those without proteinuria (n = 1,804) are presented in Table 1. More than half of patients in both groups were women. In addition, the majority were over 65 years of age. The group with proteinuria had more patients with a nursing dependency score ≥1 than that without proteinuria (62% vs. 45%, P <0.0001). There were no significant differences between patients with and without proteinuria in terms of CCI (excluding cancer and metastasis) and the following comorbidities: congestive heart failure, diabetes, and kidney disease. The incidence of colorectal cancer was the highest among all cancers (>50%), followed by that of non-small-cell lung cancer, ovarian cancer, breast cancer, cervical cancer, and malignant glioma. Although the group with proteinuria had more patients with ovarian cancer (12% vs. 8%, P <0.01), there was no significant difference in the proportions of the cancer types between the groups.

Laboratory data were compared between the patients with and without proteinuria (Table 2). SBP ≥140 mmHg was observed significantly more frequently in patients with proteinuria than in those without proteinuria (39% vs. 29%, P <0.0001). Similarly, DBP ≥90 mmHg was observed more frequently in patients with proteinuria than in those without proteinuria (20% vs. 17%, P <0.0001). Scr ≥0.68 mg/dL (median value) was also observed significantly more frequently in patients with proteinuria than in those without proteinuria (38% vs. 31%, P = 0.0094). Low eGFR and Ccr values were observed more frequently in patients with proteinuria than in those without proteinuria (P = 0.0009 and P = 0.0110, respectively). Higher values of blood urea nitrogen and total bilirubin were observed in patients with proteinuria than in those without proteinuria (P = 0.0487 and P = 0.0493, respectively).

## Predictors of proteinuria after bevacizumab administration

The results of the multivariable analysis conducted to detect the factors associated with proteinuria after bevacizumab administration are presented in Table 3. The risk of proteinuria with ovarian cancer was 1.6 times higher than that of proteinuria with colorectal cancer, based on odds ratio (OR) (95% confidence interval [CI], 1.1939–2.2561; P = 0.0023). Patients who exhibited a nursing dependency score ≥1 had greater risk of proteinuria than those with a nursing dependency score of 0 (OR, 2.3987; 95% CI, 1.8878–3.0479; P <0.0001). The risk of proteinuria in patients with SBP ≥140 mmHg was ≥ 1.4 times higher than that in patients with SBP <140 mmHg, based on OR (95% CI, 1.1671–1.7888; P = 0.0007). Moreover, the risk of proteinuria in patients with eGFR values of 60–89, 45–59, and < 45 mL/min/1.73 $m^2$ were 1.3 and 1.8 times higher than those in patients with an eGFR of >90 mL/min/1.73 $m^2$ (95% CI, 0.9294–1.8088, P = 0.1262; 95% CI, 1.0820–2.8900, P = 0.0229; and 95% CI, 0.7855–3.5080, P = 0.1843, respectively). The risk of proteinuria in patients that received RAA inhibitors on the index day was 70% higher than that in patients who did not take any antihypertensives; however, the risk was not significantly high (95% CI, 0.4924–1.0219; P = 0.0652).

## Medication utilization

Data on chemotherapy and antihypertensive drug use on the index day are presented in Table 4. Significantly more patients were administered plant alkaloid-based chemotherapy in

**Table 1. Demographic and clinical characteristics.**

| | | Proteinuria | No proteinuria | P-value |
|---|---|---|---|---|
| **Number of patients** | | 654 | 1804 | |
| **Male, n (%)** | | 272 (41.59) | 759 (42.07) | 0.8302 |
| **Age, years, n (%)** | | | | 0.2272 |
| | <65 | 275 (42.05) | 785 (43.51) | |
| | ≥65, <75 | 244 (37.31) | 702 (38.91) | |
| | ≥75 | 135 (20.64) | 317 (17.57) | |
| **Body mass index (BMI), kg/m$^2$, n (%)** | | | | 0.4028 |
| | <18.5 | 84 (12.84) | 246 (13.64) | |
| | ≥18.5, <25 | 423 (64.68) | 1197 (66.35) | |
| | ≥25 | 147 (22.48) | 361 (20.01) | |
| **Nursing dependency score[#1], sum, n (%)** | | | | <0.0001 |
| | 0 | 122 (18.65) | 540 (29.93) | |
| | ≥1 | 405 (61.93) | 819 (45.40) | |
| | Missing | 127 (19.42) | 445 (24.67) | |
| **Charlson comorbidity index[#2], sum, n (%)** | | | | 0.5372 |
| | 0 | 477 (72.94) | 1293 (71.67) | |
| | ≥1 | 177 (27.06) | 511 (28.33) | |
| **Comorbidity[#3], n (%)** | Congestive heart failure[#4] | 12 (1.83) | 33 (1.83) | 0.9927 |
| | Diabetes[#5] | 86 (13.15) | 217 (12.03) | 0.4577 |
| | Kidney disease[#6] | 28 (4.28) | 58 (3.22) | 0.2130 |
| **Type of cancer[#7], n (%)** | | | | 0.0898 |
| | Colorectal cancer | 351 (53.67) | 999 (55.38) | 0.4522 |
| | Non-small-cell lung cancer | 132 (20.18) | 418 (23.17) | 0.1163 |
| | Ovarian cancer | 77 (11.77) | 148 (8.20) | <0.01 |
| | Breast cancer | 34 (5.20) | 103 (5.71) | 0.6257 |
| | Cervical cancer | 30 (4.59) | 73 (4.05) | 0.5545 |
| | Malignant glioma | 19 (2.91) | 44 (2.44) | 0.5181 |
| | Others | 11 (1.68) | 19 (1.05) | 0.2096 |
| **Chemotherapy before the index day[#8], n (%)** | | | | 0.5907 |
| | Non-applied | 348 (53.21) | 982 (54.43) | |
| | Applied | 306 (46.79) | 822 (45.57) | |

P-values were calculated using the chi-square test. Abbreviations: ICD-10, International Statistical Classification of Diseases and Related Health Problems, 10[th] Revision.

#1: On the index day

#2: All disease names except for cancer and metastatic carcinoma

#3: Comorbidity disease name

#4: ICD-10; I43, 50, 099, 110, 130, 132, 255, 420, 425–429, and P290

#5: ICD-10; E100-149

#6: ICD-10; N00-39

#7: All disease names

#8: six months before the index day.

conjunction with bevacizumab in the group with proteinuria than in the group without proteinuria (43% vs. 38%, P = 0.0477).

There were no significant differences in antihypertensive drug use between patients with and without proteinuria. Forty-five (7%) of the 654 patients with proteinuria were administered only RAA inhibitors. Of the 1,804 patients without proteinuria, 152 (8%) were administered only RAA inhibitors. The number of patients taking other antihypertensives was 41 (6%)

**Table 2. Laboratory data.**

| | | Proteinuria | No proteinuria | P-value |
|---|---|---|---|---|
| **Number of patients** | | 654 | 1804 | |
| **SBP, mmHg** | | | | < .0001 |
| | <140 | 376 (57.49) | 1119 (62.03) | |
| | ≥140 | 257 (39.30) | 516 (28.60) | |
| | Missing | 21 (3.21) | 169 (9.37) | |
| **DBP, mmHg** | | | | < .0001 |
| | <90 | 500 (76.45) | 1331 (73.78) | |
| | ≥90 | 133 (20.34) | 304 (16.85) | |
| | Missing | 21 (3.21) | 169 (9.37) | |
| **Scr[#1], mg/dL** | | | | 0.0094 |
| | <0.68 | 196 (29.97) | 575 (31.87) | |
| | ≥0.68 | 246 (37.61) | 562 (31.15) | |
| | Missing | 212 (32.42) | 667 (36.97) | |
| **eGFR[#2], mL/min/1.73 m$^2$** | | | | 0.0009 |
| | <45 | 18 (2.75) | 33 (1.83) | |
| | ≥45, <60 | 84 (12.84) | 150 (8.31) | |
| | ≥60, <90 | 464 (70.95) | 1306 (72.39) | |
| | ≥90 | 88 (13.46) | 315 (17.46) | |
| **Ccr[#3], mL/min** | | | | 0.0110 |
| | ≤70 | 208 (31.80) | 467 (25.89) | |
| | >70 | 234 (35.78) | 667 (36.97) | |
| | Missing | 212 (32.42) | 670 (37.14) | |
| **Alb, g/dL** | | | | 0.1809 |
| | ≤3.5 | 118 (18.04) | 312 (17.29) | |
| | >3.5 | 266 (40.67) | 673 (37.31) | |
| | Missing | 270 (41.28) | 819 (45.40) | |
| **BUN, mg/dL** | | | | 0.0487 |
| | ≤20 | 391 (59.79) | 1023 (56.71) | |
| | >20 | 55 (8.46) | 120 (6.65) | |
| | Missing | 208 (31.80) | 661 (36.64) | |
| **T-Bil, mg/dL** | | | | 0.0493 |
| | <1.5 | 424 (64.83) | 1079 (59.81) | |
| | ≥1.5 | 13 (1.99) | 30 (1.66) | |
| | Missing | 217 (33.18) | 695 (38.53) | |
| **HbA1c[#4], %** | | | | 0.2394 |
| | <6.5 | 13 (1.99) | 36 (2.00) | |
| | ≥6.5 | 6 (0.92) | 33 (1.83) | |
| | Missing | 635 (97.09) | 1735 (96.18) | |

Laboratory test values measured on or one day prior to the index day were analyzed. P-values were calculated using the chi-square test. Abbreviations: SBP, systolic blood pressure; DBP, diastolic blood pressure; Scr, serum creatinine; eGFR, estimated glomerular filtration rate; Ccr, creatinine clearance; Alb, albumin; BUN, blood urea nitrogen; T-Bil, total bilirubin; HbA1c, glycated hemoglobin; N/A, not applicable

#1: The patients were categorized by median value (0.68 mg/dL)

#2: $194 \times Scr-1.094 \times Age-0.287$ ($\times 0.739$ if female). If Scr was missing, 75 mL/min/1.73 m$^2$ was used as the eGFR value.

#3: $((140-Age) \times Weight)/(72 \times Scr)$ ($\times 0.85$ if female)

#4: the most recent data from Day -29–1.

**Table 3. Predictors of proteinuria after bevacizumab administration.**

| | | OR [95% CI] | P-value |
|---|---|---|---|
| **BMI** | | | |
| | <18.5 | Ref. | |
| | ≥18.5, <25 | 1.0461 [0.7860–1.3923] | 0.7471 |
| | ≥25 | 1.1549 [0.8208–1.6251] | 0.4085 |
| **Diabetes** | | | |
| | - | Ref. | |
| | + | 1.0958 [0.8240–1.4573] | 0.5292 |
| **Type of cancer** | | | |
| | Colorectal cancer | Ref. | |
| | Non-small-cell lung cancer | 0.8885 [0.6968–1.1331] | 0.3407 |
| | Ovarian cancer | 1.6412 [1.1939–2.2561] | 0.0023 |
| | Breast cancer | 1.1289 [0.7359–1.7317] | 0.5788 |
| | Cervical cancer | 1.2435 [0.7840–1.9725] | 0.3545 |
| | Malignant glioma | 1.7679 [0.9935–3.1460] | 0.0527 |
| | Others | 1.6447 [0.7534–3.5903] | 0.2116 |
| **Nursing dependency score** | | | |
| | 0 | Ref. | |
| | ≥1 | 2.3987 [1.8878–3.0479] | < .0001 |
| **SBP** | | | |
| | <140 | Ref. | |
| | ≥140 | 1.4449 [1.1671–1.7888] | 0.0007 |
| **DBP** | | | |
| | <90 | Ref. | |
| | ≥90 | 1.0060 [0.7800–1.2976] | 0.9632 |
| **Scr, mg/dL** | | | |
| | <0.68 | Ref. | |
| | ≥0.68 | 1.0035 [0.7441–1.3534] | 0.9816 |
| **eGFR** | | | |
| | ≥90 | Ref. | |
| | ≥60, <90 | 1.2966 [0.9294–1.8088] | 0.1262 |
| | ≥45, <60 | 1.7683 [1.0820–2.8900] | 0.0229 |
| | <45 | 1.6600 [0.7855–3.5080] | 0.1843 |
| **Ccr, mL/min** | | | |
| | >70 | Ref. | |
| | ≤70 | 1.0879 [0.8384–1.4117] | 0.5261 |
| **BUN, mg/dL** | | | |
| | ≤20 | Ref. | |
| | >20 | 1.0248 [0.7043–1.4910] | 0.8983 |
| **T-Bil, mg/dL** | | | |
| | <1.5 | Ref. | |
| | ≥1.5 | 1.1727 [0.5925–2.3211] | 0.6473 |
| **Antihypertensive drug** | | | |
| | No antihypertensive drug | Ref. | |
| | RAA inhibitor only | 0.7094 [0.4924–1.0219] | 0.0652 |
| | Non-RAA only | 0.9968 [0.6762–1.4694] | 0.9870 |

(*Continued*)

**Table 3.** (Continued)

|  |  | OR [95% CI] | P-value |
|---|---|---|---|
|  | RAA inhibitor + Non-RAA | 1.1575 [0.7358–1.8210] | 0.5269 |

Odds ratios and P-values were calculated using multiple variable analysis. Abbreviations: BMI, body mass index; RAA, renin-angiotensin-aldosterone system; OR, odds ratio; CI, confidence interval; SBP, systolic blood pressure; DBP, diastolic blood pressure; Scr, serum creatinine; eGFR, estimated glomerular filtration rate; Ccr, creatinine clearance; Alb, albumin; BUN, blood urea nitrogen; T-Bil, total bilirubin

among those with proteinuria and 119 (7%) among those without proteinuria. The proportion of patients who were administered both RAA inhibitors and other antihypertensives was 5% in the group with proteinuria, which was higher than that in the group without proteinuria. The proportion of patients who were not administered antihypertensive drugs was >80% in both groups.

## Discussion

In the present study, we investigated the incidence and risk factors of bevacizumab-related proteinuria and examined the effectiveness of antihypertensive drugs in preventing proteinuria. Our results showed that approximately 27% of the patients developed proteinuria and that the factors associated with proteinuria included nursing dependence, SBP ≥140 mmHg, and low eGFR. Additionally, ovarian cancer was a risk factor for proteinuria. No significant relationship was observed between antihypertensive drugs and the occurrence of proteinuria.

In our real-world study, the proportion of patients with cancer who developed proteinuria after bevacizumab administration was higher than that reported in the package insert and a

**Table 4. Medication utilization.**

|  |  | Proteinuria | No proteinuria | P-value |
|---|---|---|---|---|
| **Number of patients with** |  | 654 | 1804 |  |
| **Bevacizumab (mg/kg), n (%)** |  |  |  | 0.8561 |
|  | <10 | 280 (42.81) | 776 (43.02) |  |
|  | ≥10, <15 | 175 (26.76) | 464 (25.72) |  |
|  | ≥15 | 199 (30.43) | 564 (31.26) |  |
| **Chemotherapy, n (%)** |  |  |  |  |
|  | Platinum-based | 437 (66.82) | 1263 (70.01) | 0.1316 |
|  | Antimetabolites-based | 370 (56.57) | 1089 (60.37) | 0.0915 |
|  | Plant alkaloid-based | 278 (42.51) | 687 (38.08) | 0.0477 |
|  | Protein kinase inhibitor-based | 16 (2.45) | 40 (2.22) | 0.7383 |
|  | Cytotoxic antibiotics-based | 9 (1.38) | 22 (1.22) | 0.7606 |
|  | Alkylating-based | 6 (0.92) | 13 (0.72) | 0.6287 |
|  | Hormone antagonist-based | 5 (0.76) | 10 (0.55) | 0.5636 |
| **Antihypertensive drug at the index day, n (%)** |  |  |  | 0.4326 |
|  | RAA inhibitor only | 45 (6.88) | 152 (8.43) |  |
|  | Other antihypertensives only | 41 (6.27) | 119 (6.60) |  |
|  | RAA inhibitor + others | 32 (4.89) | 70 (3.88) |  |
|  | No antihypertensive drug | 536 (81.96) | 1463 (81.10) |  |

P-values were calculated using the chi-square test for categorical variables and the Wilcoxon rank-sum test for continuous variables. Abbreviations: RAA, renin-angiotensin-aldosterone system

previous meta-analysis [5]. Although we excluded patients with a history of bevacizumab administration, the incidence rate was >2-fold higher than that reported in the meta-analysis. This suggests the need to closely monitor patients for proteinuria after the initial administration of bevacizumab.

Patients with ovarian cancer had significantly higher odds of exhibiting proteinuria than those with colorectal cancer. A previous study suggested that ovarian cancer is a risk factor because the bevacizumab dose used for ovarian cancer (15 mg/kg) is higher than that used for colorectal cancer (5 or 7.5 mg/kg) [7]. Our results showed no correlation between the initial dose of bevacizumab and proteinuria in the bivariate analysis; however, the dose of bevacizumab during the observation period was not surveyed in this study. Therefore, it is necessary to determine the relationship between the cumulative dose of bevacizumab and proteinuria.

In patients who exhibited nursing dependence, the odds of exhibiting proteinuria were 2.4-fold higher than those in patients who were nursing independently. There was no association between proteinuria and age, sex, BMI, CCI, or other comorbidities. We hypothesized that the presence of comorbidities, especially cardiovascular and kidney diseases, would be risk factors because the pathology of proteinuria is associated with hypertension [20]. In this study, kidney function at the administration of bevacizumab was significantly worse in patients with proteinuria than in those without proteinuria. This finding suggests that it is important to monitor kidney function to prevent or manage proteinuria when bevacizumab is used.

Patients with proteinuria had significantly higher SBP and DBP on the index day than those without proteinuria, and SBP $\geq$140 mmHg was associated with an OR of 1.44 compared with SBP <140 mmHg in the multivariable analysis. According to a previous study, bevacizumab-associated hypertension is likely to occur in patients with pre-existing hypertension [21]. Because hypertension and proteinuria are related, blood pressure should be monitored from the start of bevacizumab therapy. An eGFR lower than $\geq$90 mL/min/1.73 m$^2$ was associated with higher odds of proteinuria; however, the association of proteinuria with an eGFR <45 mL/min/1.73 m$^2$ was not significant. We did not have a sufficient number of patients in this group (with low eGFR) because we excluded patients who underwent dialysis, and bevacizumab may not have been selected for patients with renal dysfunction. This could explain the lack of significance of these results.

Because bevacizumab is administered in combination with other anticancer agents, we investigated the type of chemotherapeutic administered on the index day. Platinum-based and antimetabolite-based drugs were administered along with bevacizumab in more than half of patients. As platinum-based chemotherapy causes renal toxicity [22], we expected such therapy to influence proteinuria. However, the risk of proteinuria was not significantly higher in patients who used platinum-based agents. When platinum-based agents are used, hydration treatment is administered to prevent renal toxicity, and healthcare professionals monitor renal function [23]. Our findings suggest that patients who received platinum-based agents with bevacizumab could have been monitored adequately so that proteinuria was eventually avoided. In contrast, the proportion of patients who received plant alkaloid-based agents on the index day was significantly higher among those with proteinuria than among those without proteinuria. Taxanes are representative plant alkaloid-based drugs administered to patients with female-specific cancers such as ovarian, breast, and cervical cancers. However, the administration of bevacizumab in combination with taxanes does not influence the toxicity profile of taxanes [24, 25]. In our study, although we did not include sex in the multivariable analysis, we adjusted for the type of cancer. Therefore, patients with ovarian cancer treated with plant alkaloid-based drugs could be at risk of developing proteinuria caused by bevacizumab, although further studies are required to clarify this.

To evaluate the effectiveness of antihypertensive drugs in preventing proteinuria, we focused on their use on the index day. Although approximately 30% of patients had hypertension based on the laboratory data on the index day, approximately 80% of patients who received bevacizumab did not take any antihypertensive drugs. Among the patients who received antihypertensive drugs, approximately 65% were treated with RAA inhibitors alone or in combination with other types of antihypertensive drugs. The type of antihypertensive drug administered on the index day was not associated with proteinuria. We hypothesized that RAA inhibitors could prevent proteinuria caused by bevacizumab. However, we found no association between RAA inhibitor use on the index day and proteinuria. A previous study on the use of antihypertensives during bevacizumab therapy reported that higher doses of a single agent or more than one antihypertensive agent are required to normalize blood pressure [21]. This implies that the duration of antihypertensive use should be investigated in further studies to reveal their effect on proteinuria.

The present study had some limitations. First, we did not investigate the duration of bevacizumab and antihypertensive drug use after the index day. A previous long-term extension study of bevacizumab administration showed that adverse effects occur in 56.8% of patients with cancer, the most common of which were proteinuria (41.1%) and hypertension (10.5%) [26]. Thus, continuous use of these drugs may affect outcomes. Second, the observation period was only 12 months. Data on the incidence of proteinuria after 12 months are required to understand the long-term adverse effects. Finally, because we used clinical data, it is possible that treatment such as hydration therapy administered in conjunction with platinum-based chemotherapy ameliorates the risk of proteinuria, which may have affected our results. Lastly, because our research was a cross-sectional study, we did not conduct an evaluation of proteinuria prevention using RAA inhibitors in this case. Therefore, we cannot state whether the drug benefits these patients. Further study is required to understand the effects of RAA inhibitors on the prevention of proteinuria.

In conclusion, we found that the incidence of proteinuria among patients who were administered bevacizumab was more than two-fold higher than that described in the package insert. We also found that nursing dependence, SBP $\geq$140 mmHg, and low eGFR at the time of bevacizumab administration were risk factors for proteinuria. The type of antihypertensive drug administered after bevacizumab did not significantly affect proteinuria occurrence. These findings suggest that the incidence of proteinuria is higher than currently reported. Therefore, patients on bevacizumab may be at a higher risk of proteinuria and should be monitored carefully.

## Supporting information

**S1 Checklist. STROBE statement—checklist of items that should be included in reports of observational studies.**
(DOCX)

**S1 Table. ICD-10 codes for identification of cancer type.**
(DOCX)

**S2 Table. Variables list.**
(DOCX)

## Acknowledgments

We thank Mr. Masaya Nakadera for extracting the dataset and Mr. Masanao Igarashi for providing a secure network. We also thank Mr. Kazuki Matsui for help with the laboratory test

database and Dr. Masato Koizumi for providing information on the Anatomical Therapeutic Chemical Classification System. We also thank Editage (www.editage.jp) for English language editing.

## Author Contributions

**Conceptualization:** Anna Kiyomi, Shinobu Imai.

**Data curation:** Anna Kiyomi.

**Formal analysis:** Anna Kiyomi.

**Funding acquisition:** Anna Kiyomi.

**Investigation:** Anna Kiyomi, Fukiko Koizumi.

**Methodology:** Anna Kiyomi, Shinobu Imai, Hayato Yamana.

**Project administration:** Anna Kiyomi.

**Resources:** Anna Kiyomi, Hiromasa Horiguchi.

**Supervision:** Shinobu Imai, Hayato Yamana, Hiromasa Horiguchi, Kiyohide Fushimi, Munetoshi Sugiura.

**Validation:** Anna Kiyomi.

**Visualization:** Anna Kiyomi.

**Writing – original draft:** Anna Kiyomi.

**Writing – review & editing:** Shinobu Imai, Hayato Yamana, Kiyohide Fushimi, Munetoshi Sugiura.

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
