## [Decision Letter · Decision Letter 0]

28 Feb 2023

PONE-D-23-01693Bevacizumab-induced proteinuria and its association with antihypertensive drugs: a retrospective cohort study using a Japanese administrative databasePLOS ONE

Dear Dr Kiyomi.

Thank you for submitting your manuscript to PLOS ONE. After careful consideration, we feel that it has merit but does not fully meet PLOS ONE’s publication criteria as it currently stands. Therefore, we invite you to submit a revised version of the manuscript that addresses the points raised during the review process. Although of retrospective neture, I find this study interesting and clinically important. Please kindly try to respond to all reviewers' comments. However, if the data the reviewers ask for are not available in the administrative database you use for this study please clearly state it when responsing to the reviewers.

Please submit your revised manuscript by March 14th 2023 If you will need more time than this to complete your revisions, please reply to this message or contact the journal office at plosone@plos.org. Please include the following items when submitting your revised manuscript:A rebuttal letter that responds to each point raised by the academic editor and reviewer(s). You should upload this letter as a separate file labeled 'Response to Reviewers'.A marked-up copy of your manuscript that highlights changes made to the original version. You should upload this as a separate file labeled 'Revised Manuscript with Track Changes'.An unmarked version of your revised paper without tracked changes. You should upload this as a separate file labeled 'Manuscript'.

We look forward to receiving your revised manuscript.

Kind regards,

Mirosława Püsküllüoğlu, MD, PhD

Academic Editor

PLOS ONE

Journal Requirements:

Reviewers' comments:

Reviewer's Responses to Questions

**Comments to the Author**

1. Is the manuscript technically sound, and do the data support the conclusions?

Reviewer #1: Partly

Reviewer #2: Yes

2. Has the statistical analysis been performed appropriately and rigorously? 

Reviewer #1: Yes

Reviewer #2: Yes

3. Have the authors made all data underlying the findings in their manuscript fully available?

Reviewer #1: Yes

Reviewer #2: Yes

4. Is the manuscript presented in an intelligible fashion and written in standard English?

Reviewer #1: Yes

Reviewer #2: Yes

5. Review Comments to the Author

Reviewer #1: In this study

Age, BMI, Charlson Index, type of Cancer, previous Chemotherapy and death were not associated with protU.

Lower eGFR is associated with protU but without a direct correlation

Predictive risk factors for protU after use of bevacizumab were nursing dependency and systolic blood pressure.

It is necessary to define nursing dependency in this population; you should compare patients with and without nursing dependency. This will possibly help to clarify and identify the factors associated with development of protU in this population. The comorbidity index that could be associated with nursing dependency did not differentiate the development of protU.

Were the renin-angiotensin-aldosterone inhibitors, used to prevent proteinuria, not useful in these patients?

The Table results is difficult to read. It is presented in successive pages and the first line of new pages has no information – ProtU, noProtU and p-value, etc. should be added in every new page

Reviewer #2: Dear Authors,

I have reviewed your work titled "Bevacizumab-induced proteinuria and its association with antihypertensive drugs: a retrospective cohort study using a Japanese administrative database".

It was direct with well outlined research question and objectives.

It would be interesting to note what the relationship of the occurrence of proteinuria and renal funciotn would be if the relationship were assessed according to current CKD GFR staging 1 to 5. The authors note that the exclusion of patients on dialysis (and thus the lower patient numbers) accounted for the relationship observed between Bevacizumab and eGFR <45. However eGFR <45 encompasses a broad group of patients from CKD 3b - CKD 5 for whom proteinuria risk profiles might be different.

The authors are however correct in pointing out that there may not be enough numbers in the group of GFR < 45 due to a form of selection bias with respect to the patient who gets the medication in the first place i.e. indication bias.

As you have rightly highlighted in the limitations, the role of continued use of antihypertensives in modifying the outcome of proteinuria is one that must be borne in mind but your study could not unfortunately evaluate. As your data was from an administrative data set, I would suspect that data can be gathered for this information and incorporated into this study.

The higher occurrence of proteinuria reported in this study than that documented in the medications drug insert does raise some concerns and hopefully triggers further research and clarifications.

Thank you.

6. PLOS authors have the option to publish the peer review history of their article (what does this mean?). If published, this will include your full peer review and any attached files.

Reviewer #1: **Yes: **Teresa Adragao

Reviewer #2: No

---

## [Author Response · Author response to Decision Letter 0]

30 Mar 2023

Thank you very much for your e-mail dated March 1, 2023, regarding our manuscript titled, “Bevacizumab-induced proteinuria and its association with antihypertensive drugs: a retrospective cohort study using a Japanese administrative database,” coauthored by Anna Kiyomi, Fukiko Koizumi, Shinobu Imai, Hayato Yamana, Hiromasa Horiguchi, Kiyohide Fushimi, and Munetoshi Sugiura. The manuscript ID is PONE-D-23-01693. We appreciate the comments from the reviewers, which have greatly improved the quality of our manuscript. We have provided point-by-point responses to the reviewers’ comments, as shown below, and have addressed and clarified the reviewers’ concerns.

Regarding the data availability statement, the administrative data is well protected and has restricted access. The data was purchased from the National Hospital Organization and can only be accessed on-site. No one can bring the data outside of their secured analysis room, even though the data is anonymous. Because of this reason, the data is legally restricted. I have attached the statement of this regulation (see page 9, 8. Data Utilization, 1. Data Utilization policy, the third bullet) as evidence.

Response to reviewers

Reviewer #1: In this study, Age, BMI, Charlson Index, type of Cancer, previous Chemotherapy and death were not associated with protU. Lower eGFR is associated with protU but without a direct correlation. Predictive risk factors for protU after use of bevacizumab were nursing dependency and systolic blood pressure.

It is necessary to define nursing dependency in this population; you should compare patients with and without nursing dependency. This will possibly help to clarify and identify the factors associated with development of protU in this population. The comorbidity index that could be associated with nursing dependency did not differentiate the development of protU.

Response: 

We appreciate the reviewer for taking the time to review our manuscript and provide valuable feedback. As the reviewer mentioned below, our table was difficult to read, and we realize that it might have caused misreading regarding nursing dependency. We did indeed compare nursing dependency (with and without) among the patients with and without proteinuria (Table 1 and line 175. Moreover, patients who exhibited a nursing dependency score ≥1 had greater odds of proteinuria than those with a nursing dependency score of 0 (OR, 2.3984; 95% CI, 1.8878-3.0473; P <0.0001). This is exactly what the reviewer mentioned and is shown in Table 3. We modified the tables to make them easier to read and added the definition of nursing dependency in lines 123-125.

Were the renin-angiotensin-aldosterone inhibitors, used to prevent proteinuria, not useful in these patients?

Response: 

Thank you for pointing this out. Because our research was a cross-sectional study, we did not conduct an evaluation of proteinuria prevention using renin-angiotensin-aldosterone inhibitors in this case. Therefore, we cannot state whether the drug benefits these patients. Further study is required to understand the effects of renin-angiotensin-aldosterone inhibitors on the prevention of proteinuria. We added this content to lines 340-344.

The Table results is difficult to read. It is presented in successive pages and the first line of new pages has no information – ProtU, noProtU and p-value, etc. should be added in every new page.

Response: 

Thank you for your comment, and we apologize for the difficulty in reading the tables. We adjusted each table to make it easier to read in the revised manuscript, but we believe that the journal editor will modify the tables again when they work on the page layout.

Reviewer #2: Dear Authors,

I have reviewed your work titled "Bevacizumab-induced proteinuria and its association with antihypertensive drugs: a retrospective cohort study using a Japanese administrative database".

It was direct with well outlined research question and objectives.

It would be interesting to note what the relationship of the occurrence of proteinuria and renal function would be if the relationship were assessed according to current CKD GFR staging 1 to 5. The authors note that the exclusion of patients on dialysis (and thus the lower patient numbers) accounted for the relationship observed between Bevacizumab and eGFR <45. However eGFR <45 encompasses a broad group of patients from CKD 3b - CKD 5 for whom proteinuria risk profiles might be different.

The authors are however correct in pointing out that there may not be enough numbers in the group of GFR < 45 due to a form of selection bias with respect to the patient who gets the medication in the first place i.e. indication bias.

As you have rightly highlighted in the limitations, the role of continued use of antihypertensives in modifying the outcome of proteinuria is one that must be borne in mind, but your study could not unfortunately evaluate. As your data was from an administrative data set, I would suspect that data can be gathered for this information and incorporated into this study.

The higher occurrence of proteinuria reported in this study than that documented in the medications drug insert does raise some concerns and hopefully triggers further research and clarifications.

Response:

We appreciate the reviewer for taking the time to review our manuscript and provide valuable feedback. As the reviewer mentioned, we categorized the patients based on their renal function to conduct the analysis. Although there were some limitations in this study, we revealed real-world evidence that proteinuria occurrence in patients who were administrated bevacizumab was higher than what the clinical study showed. Likewise, as the reviewer mentioned, we believe this finding could be beneficial in clinical practice for patients administrated bevacizumab and that further studies detecting the prevention of proteinuria by the renin-angiotensin-aldosterone inhibitors should be conducted. Therefore, based on the reviewer’s comments, we feel no additional changes are needed.

Thank you again to the reviewers and editorial team for reviewing our manuscript and providing constructive feedback regarding our research. The modified portions are colored in red in the revised manuscript. We believe that the revised manuscript, which addresses the reviewer’s concerns, is now ready for publication in PLOS ONE.

Thank you for your time, and we look forward to hearing your decision soon.

---

## [Decision Letter · Decision Letter 1]

4 Jun 2023

PONE-D-23-01693R1Bevacizumab-induced proteinuria and its association with antihypertensive drugs: a retrospective cohort study using a Japanese administrative databasePLOS ONE

Dear Dr. Kiyomi,

Thank you for submitting your manuscript to PLOS ONE. After careful consideration, we feel that it has merit but does not fully meet PLOS ONE’s publication criteria as it currently stands. Therefore, we invite you to submit a revised version of the manuscript that addresses the points raised during the review process.

We look forward to receiving your revised manuscript.

Kind regards,

Mirosława Püsküllüoğlu, MD, PhD

Academic Editor

PLOS ONE

Journal Requirements:

Reviewers' comments:

Reviewer's Responses to Questions

**Comments to the Author**

1. If the authors have adequately addressed your comments raised in a previous round of review and you feel that this manuscript is now acceptable for publication, you may indicate that here to bypass the “Comments to the Author” section, enter your conflict of interest statement in the “Confidential to Editor” section, and submit your "Accept" recommendation.

Reviewer #2: All comments have been addressed

Reviewer #3: (No Response)

2. Is the manuscript technically sound, and do the data support the conclusions?

Reviewer #2: Yes

Reviewer #3: Yes

3. Has the statistical analysis been performed appropriately and rigorously? 

Reviewer #2: Yes

Reviewer #3: No

4. Have the authors made all data underlying the findings in their manuscript fully available?

Reviewer #2: Yes

Reviewer #3: Yes

5. Is the manuscript presented in an intelligible fashion and written in standard English?

Reviewer #2: Yes

Reviewer #3: Yes

6. Review Comments to the Author

Reviewer #2: Dear Authors,

Thank you for addressing all concerns raised at the initial review of this manuscript.

Reviewer #3: Dear Authors,

I have thoroughly reviewed your paper and find it to be highly interesting and valuable for clinical practice. The clarity, transparency, conciseness, and readability of the text are commendable. However, I have some suggestions that I believe will further enhance the scientific robustness and readiness of this article.

Language: The English language usage in the paper is understandable and flows smoothly. I have no specific remarks in this regard.

Major remark: The core findings of the paper rely on subgroup comparisons and univariate analysis. Considering that proteinuria can be influenced by various factors, such as hypertension, diabetes, and obesity (Ong et al., Nephrology (Carlton), 2013), it is advisable to standardize the presented data to account for these factors and eliminate their influence on the established results. I recommend consulting with a statistician and considering the incorporation of multivariate analysis with standardization to the aforementioned factors.

Minor remarks:

Line 82: It would be beneficial to provide an explanation for why weight and height values were considered exclusion criteria.

Line 109-110: Could you please clarify whether the proteinuria values mentioned are equal? If so, providing a reference for these information would be helpful.

Line 129: It would be beneficial to specify the chemotherapeutic agents coded in the Anatomical Therapeutic Chemical (ATC) classification. Are they the same as those listed in Table 4?

Line 164-165: The information provided here is somewhat unclear and may not be necessary. If you intended to convey that during observation period, 15.1% of patients with proteinuria and 15.2% of patients without proteinuria died, it is important to provide more precise details or at least specify the time period during which these deaths occurred and the reasons behind them. Otherwise, readers may mistakenly assume that proteinuria was the direct cause of death in these individuals.

Table 1: Is it possible to include p-values for the different types of cancers rather than providing only general information for all diseases? I would suggest replacing the +/- symbols in the "chemotherapy before the index day" row with descriptive terms such as "Applied" and "Non-applied." Additionally, since readers may not be familiar with the International Classification of Diseases (ICD) codes, it would be helpful to provide their specific definitions in a footnote.

Regarding lines 212-225, as detailed data are already presented in Table 3, I recommend avoiding repetition and transforming this paragraph into a more narrative and descriptive form. For example, you could state that XYZ increases the risk of proteinuria by 20%.

Overall, I commend your work and believe that addressing these suggestions will further enhance the quality and impact of your research. I look forward to seeing the revised version of your paper.

7. PLOS authors have the option to publish the peer review history of their article (what does this mean?). If published, this will include your full peer review and any attached files.

Reviewer #2: No

Reviewer #3: **Yes: **Renata Pacholczak-Madej

---

## [Author Response · Author response to Decision Letter 1]

14 Jul 2023

TO: Dr. Mirosława Püsküllüoğlu, MD, PhD., Academic Editor, PLOS ONE

RE: Resubmission of manuscript [PONE-D-23-01693]

DATE: July 15, 2023

Dear Dr. Mirosława Püsküllüoğlu:

Thank you for your e-mail dated June 5, 2023, regarding our manuscript titled, “Bevacizumab-induced proteinuria and its association with antihypertensive drugs: A retrospective cohort study using a Japanese administrative database,” coauthored by Anna Kiyomi, Fukiko Koizumi, Shinobu Imai, Hayato Yamana, Hiromasa Horiguchi, Kiyohide Fushimi, and Munetoshi Sugiura. The manuscript ID is PONE-D-23-01693. 

We appreciate the comments from the reviewers, which have greatly improved the quality of our manuscript. The manuscript has been rechecked and the necessary changes have been made in accordance with the reviewers’ suggestions. We have clarified all the concerns of the reviewers and have provided point-by-point responses to all the comments, as shown below.

Response to reviewers

Reviewer #3: 

Major remark: The core findings of the paper rely on subgroup comparisons and univariate analysis. Considering that proteinuria can be influenced by various factors, such as hypertension, diabetes, and obesity (Ong et al., Nephrology (Carlton), 2013), it is advisable to standardize the presented data to account for these factors and eliminate their influence on the established results. I recommend consulting with a statistician and considering the incorporation of multivariate analysis with standardization to the aforementioned factors.

Response: 

We appreciate the reviewer for taking the time to review our manuscript and for providing valuable feedback. As the reviewer mentioned, several factors, particularly obesity and diabetes, could have effects on proteinuria. As we already used the variables of blood pressure, we have added obesity (BMI) and diabetes to the multivariate logistic regression analysis. The methods and results are presented in lines 153, 218�232, and Table 3. Following the additional analyses, our results did not change.

Minor remarks:

Line 82: It would be beneficial to provide an explanation for why weight and height values were considered exclusion criteria.

Response: 

Thank you for your suggestion. We have excluded the patients with weight of <25 kg or height of <100 cm; they are considered to have been registered incorrectly, with values such as 0 or to have switched weight and height values. We have added the information in lines 82�84.

Line 109-110: Could you please clarify whether the proteinuria values mentioned are equal? If so, providing a reference for this information would be helpful.

Response: Thank you for your suggestion. Although these values are not necessarily the same, the Common Terminology Criteria for Adverse Events (CTCAE) v5.0 cites the definition and proteinuria positivity values. We have inserted this information in lines 112�113.

Line 129: It would be beneficial to specify the chemotherapeutic agents coded in the Anatomical Therapeutic Chemical (ATC) classification. Are they the same as those listed in Table 4?

Response: Thank you for your suggestion. As the reviewer mentioned, the chemotherapeutic agents coded in the ATC classification are the same as those listed in Table 4. Therefore, we have added this content in lines 130�132 to specify the codes; L01A: alkylating-based, L01B: antimetabolites-based, L01C: plant alkaloid-based, L01D: cytotoxic antibiotics-based, L01E: protein kinase inhibitor-based, L01X: platinum-based, and L02B: hormone antagonist-based agents.

Line 164-165: The information provided here is somewhat unclear and may not be necessary. If you intended to convey that during observation period, 15.1% of patients with proteinuria and 15.2% of patients without proteinuria died, it is important to provide more precise details or at least specify the time period during which these deaths occurred and the reasons behind them. Otherwise, readers may mistakenly assume that proteinuria was the direct cause of death in these individuals.

Response: Thank you for your valuable suggestion. As the reviewer suggested, because this study was conducted as a cross-sectional study, it is impossible to discuss further about why and how the death occurred. Consequently, we have removed the content from the revised manuscript.

Table 1: Is it possible to include p-values for the different types of cancers rather than providing only general information for all diseases? I would suggest replacing the +/- symbols in the "chemotherapy before the index day" row with descriptive terms such as "Applied" and "Non-applied." Additionally, since readers may not be familiar with the International Classification of Diseases (ICD) codes, it would be helpful to provide their specific definitions in a footnote.

Response: Considering the reviewer’s comment, we have added each p-value for the different types of cancers and modified the results (lines 182–184). The symbols regarding chemotherapy in Table 1 were changed to “Applied” and “Non-applied.” We have also included the definitions of the ICD code in a footnote.

Regarding lines 212-225, as detailed data are already presented in Table 3, I recommend avoiding repetition and transforming this paragraph into a more narrative and descriptive form. For example, you could state that XYZ increases the risk of proteinuria by 20%.

Response: 

Thank you for your suggestion. According to the reviewer’s recommendations, we have modified the explanation to make them easier to understand in lines 219–232.

Thank you once again to the reviewers and editorial team for reviewing our manuscript and providing constructive feedback regarding our research. The modified portions are colored in red in the revised manuscript. We believe that the revised manuscript addresses the reviewers’ concerns and is now ready for publication in PLOS ONE.

Thank you for your time, and we look forward to hearing your decision soon. 

Sincerely,

Anna Kiyomi, PhD.

Assistant Professor

Tokyo University of Pharmacy and Life Sciences

1432-1 Horinouchi, Hachioji

Tokyo 192-0392, Japan

Tel/Fax: +81-42-676-6622

akiyomi@toyaku.ac.jp

---

## [Editor Report · Decision Letter 2]

31 Jul 2023

Bevacizumab-induced proteinuria and its association with antihypertensive drugs: A retrospective cohort study using a Japanese administrative database

PONE-D-23-01693R2

Dear Dr. Kiyomi,

We’re pleased to inform you that your manuscript has been judged scientifically suitable for publication and will be formally accepted for publication once it meets all outstanding technical requirements.

Kind regards,

Mirosława Püsküllüoğlu, MD, PhD

Academic Editor

PLOS ONE
---

## [Editor Report · Acceptance letter]

3 Aug 2023

PONE-D-23-01693R2 

Bevacizumab-induced proteinuria and its association with antihypertensive drugs: A retrospective cohort study using a Japanese administrative database 

Dear Dr. Kiyomi:

I'm pleased to inform you that your manuscript has been deemed suitable for publication in PLOS ONE. Congratulations! Your manuscript is now with our production department. 

Kind regards, 

on behalf of

Dr. Mirosława Püsküllüoğlu 

Academic Editor

PLOS ONE